# Understanding the Task and Data Misconceptions in Online Map Based Motion Prediction for Autonomous Driving and a Boundary-Free Baseline

## Abstract

In autonomous driving (AD), online high-definition (HD) map estimation is gaining increasing attention. To examine how online-estimated HD maps impact downstream tasks, the protocol of online mapping based motion prediction emerges. This protocol follows a two-stage training paradigm: online mapping models are firstly trained and then used to output map elements which are fed as inputs for motion prediction models. In this paper, we conduct in-depth study to investigate the challenges and misunderstandings associated with the protocol and propose **OMMP-Bench**, a well-defined and insightful benchmark of online map based motion prediction. We identify that the current dataset splits are unsuitable for two-stage training, leading to a severe train-validation gap, and thus we design a new data partitioning split. Furthermore, we find that the perception range of map prediction models does not fully meet the requirements of motion prediction, resulting in a lack of map elements for agents far from the ego vehicle. This issue is obscured by incorrect metrics that evaluate only the ego vehicle's trajectory. We address it by refining the metrics to evaluate all moving vehicles and separately report performance for agents under different distance ranges. Further, to alleviate the issue of missing map elements for faraway agents, we introduce a new baseline that directly uses image features generated by the online mapping model. These features are not constrained by perception range and could supplement environmental information around agents beyond the online map's coverage. We further explore how different map elements influence motion prediction, as existing online mapping models have different designs of output format. We conduct thorough experiments to verify the proposed corrections and will open source the related code and checkpoints. We hope OMMP-Bench could solve the long-standing mis-usage and misunderstanding of the emerging field and provide insights for further co-development of online mapping and motion prediction models.

## 1 Introduction

Motion prediction plays a critical role in an autonomous driving system, which forecasts the future movements of surrounding agents based on their past trajectories as well as map elements. Traditional motion prediction methods Zhou et al. (2022); Shi et al. (2022); Gu et al. (2021), as in Fig. 2 (a), assume the availability of high-definition (HD) maps with rich static environmental information, such as lane markings, centerlines, and crosswalks. Such information aids in predicting agent movement Gao et al. (2020b).

However, HD map creation is expensive due to the need for extensive data collection and manual annotation, with updates required every 2-3 months Li et al. (2022). This limits the use of methods that rely on HD maps. To mitigate this, online mapping models Liao et al. (2023a); Li et al. (2024); Yuan et al. (2024) have gained significant attention in recent days. These models utilize raw sensor data (e.g. from cameras and LiDAR) to generate HD maps of the vehicle's surroundings in real time, providing essential information for motion prediction (and other downstream tasks). Online

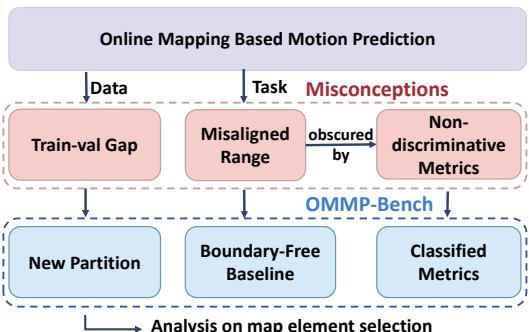

Figure 1: **Overview of Our Study.**

maps reduce the reliance on high-definition map annotations. However, the errors may impact the performance of downstream tasks such as motion prediction.

To analyze the impact of using online maps on motion prediction and promote the co-development of the two fields, the online map based motion prediction protocol emerges in 2024 Gu et al. (2024a) (CVPR 2024 Best Paper Final List), as shown in Fig. 2 (c), which attracts keen attention from the research community. It is formulated by using online maps (including features generated by online mapping models) and agents' ground truth history trajectories as inputs to the motion prediction model. Pioneering works Gu et al. (2024a;b) all adopt a two-stage paradigm. At the first stage, sensor data is used to train an online mapping model Liao et al. (2023a); Li et al. (2024); Yuan et al. (2024). Then, in the second stage, the online mapping model generates inference results for the map. This predicted map, along with historical trajectory data, is then used to train a motion prediction model. As a brand new paradigm, it is at an early stage. In this paper, **we delve deep into it and identify the following misunderstandings and challenges:**

**Inappropriate Dataset Splits.** The official splits of typical autonomous driving datasets such as nuScenes Caesar et al. (2019) are composed of training, validation, and test sets, a standard way for regular machine learning problems. However, under the two-stage training online map based motion prediction, there are two issues: (1) The online mapping model is trained on the training set first. Then, to set up the training data for the motion prediction model, the online mapping model would infer on the training set under existing protocol Gu et al. (2024a). **The predicted map would be very accurate since the online mapping model is trained on this set.** However, this is not the case during validation where the motion prediction model would take the online mapping model's prediction on the validation set as inputs. At the validation set, the online mapping model has much lower accuracy and thus introduces a huge **train-val gap for the downstream motion prediction model**. (2) Additionally, as pointed out in Yuan et al. (2024), there are **spatial overlaps among the training and validation set** of nuScenes. As a result, the official train-val split could not fully reflect the generalization ability of the online mapping models. To alleviate these two issues, we design a new partition for OMMP-Bench by cutting nuScenes into three parts without any geometry overlap: a map training set, a motion training set, and a motion validation set. The online mapping model is trained on the map training set and generates online maps on the motion training set and motion validation set.

**Different Considered Range for Online Mapping and Motion Prediction.** Popular online mapping models, such as MapTR/MapTRv2 Liao et al. (2023a;b) used in the protocol of Gu et al. (2024a), have rather small pre-defined range (for example, $\pm 15 \times \pm 30$ meters) due to the difficulty of prediction for distant map elements. However, for the motion prediction part, the nearest agent could be more than 100 meters away from the ego vehicle. As a result, **there are a bunch of agents that do not have map elements nearby**, which could significantly degenerate the accuracy of prediction Gao et al. (2020b). The pioneering protocol Gu et al. (2024a;b) sidesteps this issue by only calculating the motion prediction accuracy of the ego vehicle during evaluation. However, motion prediction is mainly designed to avoid potential collision with other agents, which the existing protocol fails to evaluate at all. To this end, in OMMP-Bench, we propose to only evaluate non-ego agents. Further, to compensate for those agents without any context, we propose a simple yet effective baseline that leverages the image feature around those agents as environmental information to address the out-of-map issue.

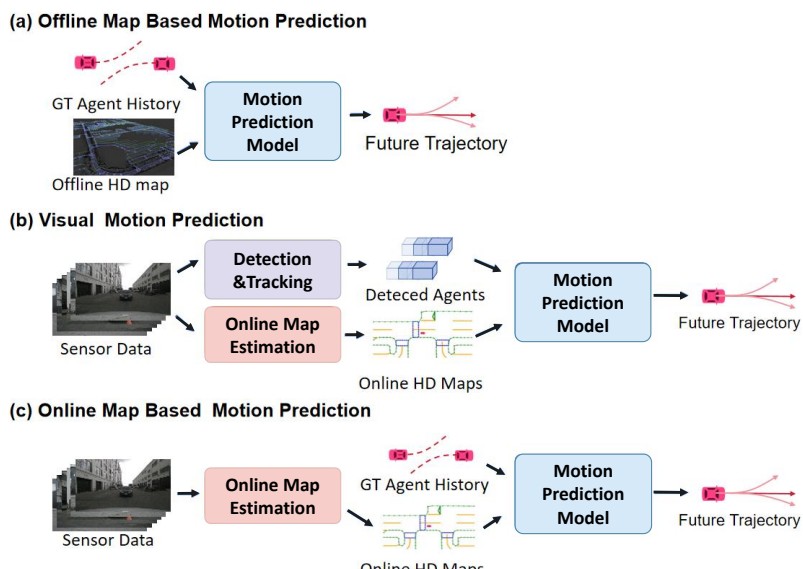

Figure 2: **Paradigm Comparison for Different Settings to Study Motion Prediction.**

**Non-discriminative Metrics.** For the motion prediction protocol Gu et al. (2024a) in nuScenes, we observe that there are lots of static agents and all motion models could perfectly predict static trajectories for these agents. As a result, the large number of easy cases would cause the metrics less distinguishable, and thus we propose to only evaluate non-static agents, similar to the design of Argoverse Lambert & Hays (2021); Wilson et al. (2021) and Waymo Ettinger et al. (2021). Further, since agents within and outside the perception range of online mapping models have rather different inputs, we separately report the motion prediction metrics of agents nearby or far away from the ego vehicle to evaluate motion prediction models' accuracy under different conditions.

**Formulation of Online Map.** As the output formulations of online mapping Liao et al. (2023a;b); Li et al. (2023) are still open problems, we investigate the influence of different existing formulations for motion prediction models on OMMP-Bench to provide insights for new designs in the online mapping community.

Notably, there is another line of study taking a step further where the agent and map inputs for motion prediction are both inference results of upstream modules, like ViP3D Gu et al. (2023) and PIP Jiang et al. (2022), as shown in Fig. 2 (b). This line of research contributes important insights regarding the recent emergence of end-to-end autonomous driving. However, the study of online mapping's influence on motion prediction perspective introduces additional complexity by coupling motion prediction performance with that of object detection and tracking. For example, if a vehicle is not detected, its trajectory cannot even be predicted. To avoid misconception and offer clear insights, in this work, we focus on the online map based motion prediction setting.

Our contributions are summarized below:

- We discover the misconceptions of data and tasks in online mapping based motion prediction, including inappropriate dataset splits, different considered ranges, and non-discriminative metrics.
- We propose a well-defined benchmark OMMP-Bench with a new partition and separate evaluation within and outside situations to avoid misleading conclusions.
- We introduce a boundary-free baseline that utilizes image features to mitigate the unaligned range issue between the online mapping and motion prediction model.
- We evaluate existing methods on OMMP-Bench and analyze the effect of map element selection on motion prediction models.

## 2 RELATED WORK

**Online Map Estimation.** Online HD map estimation methods generate HD maps from sensor data such as cameras and LiDAR. Early models Zhou & Krähenbühl (2022) approached online

map estimation as a BEV segmentation. These models first generated BEV features from sensor data using 2D-BEV transformations, then predicted the semantic category of each BEV grid cell, producing a gridded HD map. However, gridded maps lack continuity and instance information. Consequently, recent online map prediction models tend to generate vectorized maps. HDMapNet Li et al. (2022) extracts vectorized maps from gridded maps using complex post-processing, while VectorMapNet Liu et al. (2023) is the first model to directly generate vectorized maps in an end-to-end manner. MapTR Liao et al. (2023a;b) introduces a unified permutation-equivalent modeling method to accurately represent point groups, and StreamMapNet Yuan et al. (2024) leverages temporal information for map prediction. LanSegNet Li et al. (2024) introduces the concept of lane segments, combining map element detection with centerline perception.

**Motion Prediction.** Pioneering deep learning based motion prediction methods opt to use rasterized images to represent the scene Tang & Salakhutdinov (2019); Lee et al. (2017); Zhao et al. (2019); Chai et al. (2020); Cui et al. (2019); Hong et al. (2019). These approaches project map elements onto a top-down image based on 2D coordinates, with different elements painted on separate channels, allowing convolutional neural networks (CNNs) to extract map information. Later, vector-based approaches for map encoding have shown superior performance in multiple challenges. These methods typically employ 1D CNNs or LSTMs Hochreiter & Schmidhuber (1997) for temporal data, PointNet Qi et al. (2017) to process polylines, and GNNs to handle interactions between agents and map elements. VectorNet Gao et al. (2020a) uses sub-graphs for lane and agent encoding, followed by a global fully-connected graph to capture relationships. SceneTransformer Ngiam et al. (2021) introduces a factorized spatio-temporal network, applying Transformers to a fully-connected spatial/temporal graph alternately. LaneGCN Liang et al. (2020) employs four modules for aggregating scene information, while TPCN Ye et al. (2021) treats coordinate data as point clouds. Recently, HiVT and MTR Shi et al. (2022) adopt pure Transformer architecture. Their following works QCNet Zhou et al. (2023) and MTR++ Shi et al. (2023) introduce ego-centric representation Jia et al. (2022) to enhance performance.

**Online Map Based Motion Prediction.** To study the influence of online mapping on motion prediction, pioneering work MapUncertaintyPrediction Gu et al. (2024a) sets up the first protocol in the field and conveys potential error information by predicting the uncertainty of each point in the predicted vectorized map. Following work MapBEVPrediction Gu et al. (2024b) incorporates BEV features from the online map prediction model into the motion prediction model, eliminating the need for the decoder in the map model and the encoder for maps in the motion prediction model during inference. However, as mentioned above, the newly proposed protocol has several issues and we aim to overcome them and provide a clearer protocol and benchmark for the field. There are more advanced end-to-end models like ViP3D Gu et al. (2023), PIP Jiang et al. (2022), UniAD Hu et al. (2023), and VAD Jiang et al. (2023), which further make the agent input of motion prediction be the results of an upstream module as well. However, it makes the motion prediction performance deeply entangled with object detection accuracy so it is difficult to analyze the actual influence caused by the errors of online mapping.

## 3 ONLINE MAPPING BASED MOTION PREDICTION

### 3.1 PRELIMINARY

To study the influence of online mapping on motion prediction, following the pioneering protocol Gu et al. (2024a;b), there are two stages. At the first stage, sensor data is used to train an online mapping model Liao et al. (2023a); Li et al. (2024); Yuan et al. (2024). Then, in the second stage, the online mapping model generates inference results. This predicted map, along with historical trajectory data, is then used to train a motion prediction model. **By studying the performance of the final motion prediction model, it could provide insights on (1) the designs of online mapping models to better fit the requirements of motion prediction and (2) the designs of motion prediction to be more robust to imperfect online maps.** Note that all existing online mapping based motion prediction models are conducted only on nuScenes (Caesar et al., 2019) dataset because it provides raw camera data, HD maps, and trajectories of agents in the same scenario while others such as Waymo motion (Ettinger et al., 2021) or Argoverse2 motion (Wilson et al., 2021) do not. In the following sections, we delve into the details of the existing protocol of online map based motion prediction and discuss its misconceptions and our corresponding solutions in our proposed OMMP-Bench.

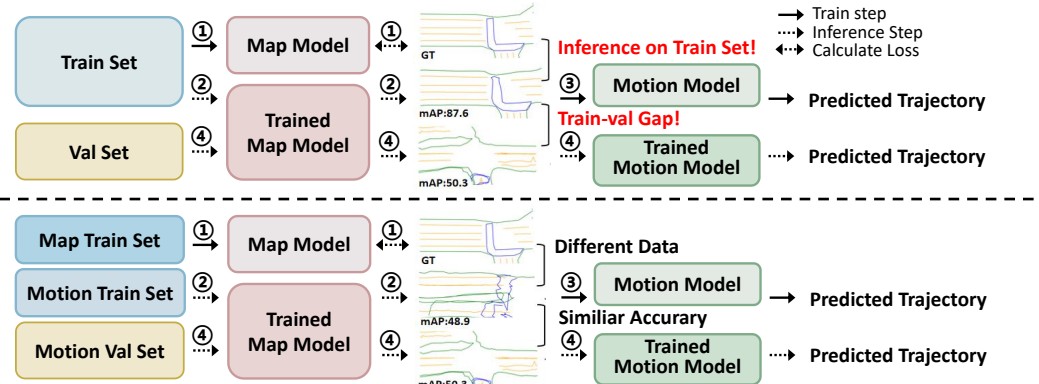

Figure 3: **Comparison of Default (Upper) and Proposed (Lower) Protocol.** Four standard steps of online mapping based motion prediction: ① training the map model; ② using the trained map model to generate maps to train the motion model; ③ training the motion model; ④ conduct inference with the trained map and motion model. Under the default setting in Gu et al. (2024a) (Upper), it would introduce train-val gaps for the motion model due to the huge difference in map accuracy. Under the proposed setting (Lower), the motion model could use maps of similar accuracy during both training and evaluation.

Table 1: **Comparison of Different Splits with MapTRv2-CL and HiVT**. The metrics are calculated on all moving non-ego vehicles.

| Split Setting | Train Map Model | Train Motion Model | Evaluation | minADE↓ | minFDE↓ | MR↓ |
|---|---|---|---|---|---|---|
| 1 (Ours) | Map Train | Motion Train | Motion Val | 0.6308 | 1.2487 | 0.1558 |
| 2 | Map Train + Motion Train | Map Train + Motion Train | Motion Val | 0.7006 | 1.3501 | 0.1817 |
| 3 (Default) Gu et al. (2024a) | nuScenes Train | nuScenes Train | nuScenes Val | 0.6839 | 1.3362 | 0.1732 |
| 4 | nuScenes Train (Sub 50%) | nuScenes Train (Another Sub 50%) | nuScenes Val | 0.6373 | 1.2261 | 0.1580 |

## 3.2 INAPPROPRIATE DATA SPLITS

In classic machine learning problems, one commonly used data partition strategy is training, validation, and test set, which is widely adopted by existing autonomous driving datasets Caesar et al. (2019); Wilson et al. (2021); Ettinger et al. (2021). However, for the task of online map based motion prediction, we find that the default split adopted in Gu et al. (2024a) is unsuitable under the two-stage training pipeline. Specifically, under the official training and validation split of nuScenesCaesar et al. (2019), the protocol would be (1) Training the online mapping model on the sensor data of the training set; (2) Using the online mapping model to inference on the training set to generate online maps; (3) Training the motion prediction model with the online maps on the training set; (4) Evaluate the motion prediction performance on the validation set with online mapping model inference on validation set first, as shown in Fig. 3 (Upper). As a result, in the training data of motion prediction, the online maps are highly accurate since the online mapping model is inference on its training set. However, during evaluation, the online mapping model encounters data it has not seen before, resulting in a significant drop in map accuracy and a shift in the distribution. **This distribution shift greatly degenerates the performance of the motion prediction model**.

Another issue of nuScenes data split has been pointed out in recent online mapping work Yuan et al. (2024). Since map elements are rather static, online mapping models could perform well on the locations covered by the training set, even with different sensor data. Unfortunately, **there exist large spatial overlaps between the training and validation set** Yuan et al. (2024) of nuScenes, as their data partition strategy divides data based on different driving logs, shown in Fig. 4 (Upper). As a result, the default split would overestimate the generalization ability of online mapping models.

To address the aforementioned issue, **we manually check the whole dataset and split it into three spatially disjoint sets for OMMP-Bench**: map train set, motion train set, and motion val set, as shown in Fig. 4 (Lower). Under this setting, the online map based motion prediction protocol is executed as: (1) training the online mapping model on the map train set; (2) using the trained map model to generate maps on the motion train set; (3) training the motion prediction model on the motion train set; (4) evaluate the motion prediction on the motion val set with online mapping model inference on the motion val set first, as shown in Fig. 3 (Upper). In this way, we make sure **the online mapping model has never encountered data from the two motion related subsets** and thus the accuracy of online maps is similar. As a result, the motion prediction model adapts to the less precise

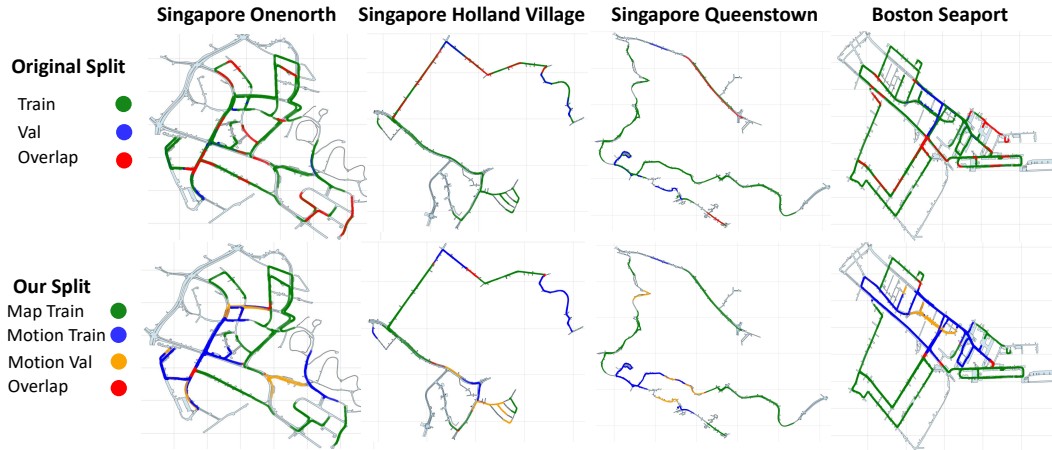

Figure 4: **Comparison of Default and Proposed Data Splits.** In default split Gu et al. (2024a), 87% of the validation data has overlap with training set. In the proposed split, only 5% of the motion train data has overlap with map train data.

Table 2: **Online Mapping Performance (mAP) under Different Perception Range.**

| Map model | 30x60m | 100x100m |
|---|---|---|
| MapTR | 0.124 | 0.014 |
| MapTRv2-CL | 0.164 | 0.002 |

Table 3: **Motion Prediction Performance of HiVT under GT Maps with Different Range.**

| Range | minADE↓ | minFDE↓ | MR↓ |
|---|---|---|---|
| 30x60m | 0.6154 | 1.2382 | 0.1448 |
| 100x100m | **0.6003** | **1.2243** | **0.1432** |

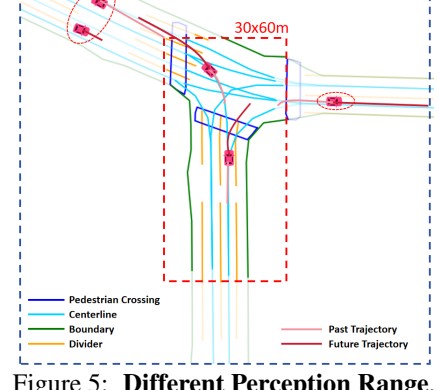

Figure 5: **Different Perception Range**.

maps, eliminating the train-validation gap. As shown in Table 1, the split of OMMP-Bench leads to an explicit performance enhancement compared to the default split, demonstrating the importance of reducing the train-val gap.

## 3.3 DIFFERENT CONSIDERED RANGE FOR ONLINE MAPPING AND MOTION PREDICTION

Autonomous driving, as an outdoor task, naturally requires defining a range of considerations to avoid unbound and unnecessary computation. Currently, most **online mapping models Liao et al. (2023a;b); Li et al. (2023) set a rather limited perception range**, due to the inherent difficulty of detecting a line in long distance with cameras. For example, MapTR only covers a 30x60m area ($\pm15 \times \pm30$m). As shown in Table 2, when extending the online mapping model to a longer range, the perception performance would drastically degenerate.

On the other hand, motion prediction tasks usually consider agents as far as more than 100 meters away from the ego vehicle in nuScenes. As a result, **a large portion of the predicted agents does not have map elements around them**, which means no surrounding information and causes degenerated performance. In Table 3 and Fig. 6, we could conclude that the absence of map elements can impact motion prediction for other vehicles, and simply expanding the perception range of the map prediction model leads to decreased map accuracy, ultimately failing to improve motion prediction performance. This indicates that current online map prediction models cannot fully meet the perception range requirements of downstream motion prediction tasks.

To enable those distant agents to obtain environment information, we propose a new baseline in OMMP-Bench to allow all agents to extract features from their corresponding nearby regions of raw image features. In this way, image features have the benefit that it does not have out-of-scope issues unlike BEV features Gu et al. (2024b). As shown in Fig. 7, We implement the integration

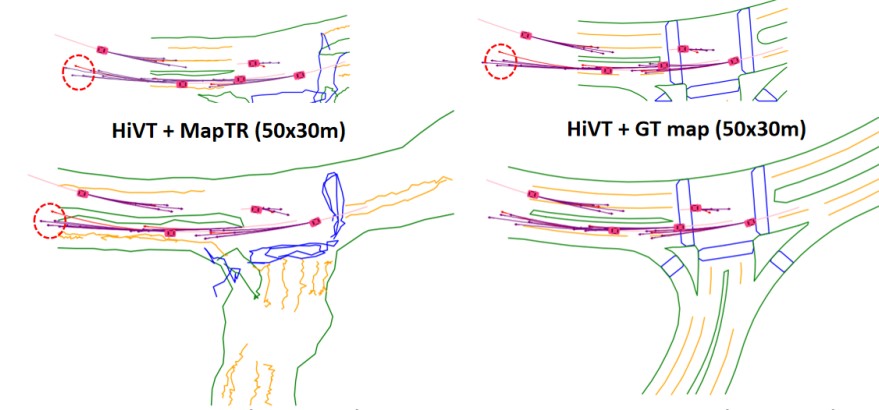

Figure 6: **Visualization of Performance under Different Perception Range.** Using long-range GT map could give explicit guidance for the far away agents while using long-range online map is not helpful.

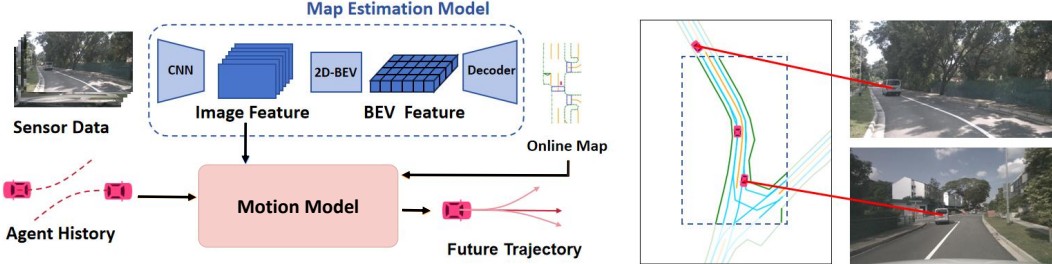

Figure 7: **Proposed Baselines to Alleviate Out-of-Scope Issue (Left). Illustration of Out-of-Scope Agents. (Right)** Applying Deformable Attention for all agents to retrieve image features enables each agent to have its own environment information.

of agent and image features using Deformable Attention. We donate the image features extracted from a backbone $\{I_1, I_2, \ldots, I_{N_c}\} \in \mathbb{R}^{H \times W}$, where (H,W) is the size of image features and $N_c$ is the number of multi-view images. Using the intrinsic and extrinsic parameters and agent positions, we project each agent onto an image feature T(i). For the i-th agent projected onto image feature T(i) at location $\mathbb{R}^2$, its feature is denoted by $A_i \in \mathbb{R}^D$.

The aggregated features are then computed as follows:

$$\hat{A}_i = \text{DeformAtt}(A_i, p_i, I_{T(i)}) \tag{1}$$

As shown in Table 4, this straightforward yet effective baseline addresses the issue of agents extending beyond the map's perceptual boundaries and achieves SOTA performance.

### 3.4 NON-DISCRIMINATIVE METRICS

The major purpose of motion prediction is to provide the intentions of surrounding agents so that the planning module can avoid collision. However, **under existing protocol, only ego vehicle's motion prediction performance is reported during evaluation**, which is against the purpose of motion prediction. Thus, in OMMP-Bench **we propose to predict other agents' future trajectories**.

Further, we observe that nuScenes contains a substantial number of stationary or slow-moving vehicles, whose motion is relatively easy for models to predict while fast-moving agents are generally more challenging to predict. Therefore, **we focused our evaluation on moving vehicles**, similar to popular motion prediction benchmarks like Argoverse Wilson et al. (2021) and Waymo Ettinger et al. (2021).

Figure 8: **Comparison of Evaluation Agent Selection Strategy.** Red vehicles represent the selected agents to report motion prediction performance about. (Left) Existing protocol Gu et al. (2024a;b) only reports results on the ego vehicle. (Right) The proposed protocol evaluates the motion prediction performance of ego vehicle and other moving agents.

Table 4: **Comparison of Online Map Based Motion Prediction.**

| Base model | Method | minADE↓ | minFDE↓ | MR↓ |
|---|---|---|---|---|
| HiVT+MapTR | base | 0.6375 | 1.2592 | 0.1585 |
| HiVT+MapTR | unc Gu et al. (2024a) | 0.6272 | 1.2487 | 0.1578 |
| HiVT+MapTR | bev Gu et al. (2024b) | 0.6287 | 1.2364 | 0.1566 |
| HiVT+MapTR | img(ours) | **0.6163** | **1.2344** | **0.1519** |

Table 5: **Performance of HiVT under Different Online Map Element Types.**

| Divider | Boundary | Ped. crossing | Centerline | minADE |
|---|---|---|---|---|
| ✓ | ✗ | ✗ | ✗ | 0.8770 |
| ✗ | ✓ | ✗ | ✗ | 0.6829 |
| ✓ | ✓ | ✗ | ✗ | 0.6558 |
| ✓ | ✓ | ✓ | ✗ | 0.6500 |
| ✗ | ✗ | ✗ | ✓ | 0.6631 |
| ✓ | ✓ | ✓ | ✓ | 0.6308 |

Table 6: **Motion Prediction Performance (minADE ↓) of Different Groups of Agents.**

| Models | Ego | All | Moving | Static | Moving Non-Ego Close | Moving Non-Ego Far |
|---|---|---|---|---|---|---|
| HiVT+MapTR | 0.4015 | 0.2224 | 0.6307 | 0.002 | 0.5585 | 0.6997 |
| DenseTNT+MapTR | 1.2114 | 0.9732 | 2.3069 | 0.009 | 2.0214 | 2.4140 |

We classify an agent as moving if it moves more than two meters within three seconds.

As discussed in Sec 3.3, some agents might fall outside of the online mapping range. To measure and observe this influence, we further divide agents into two groups: Moving-Non-Ego-Close and Moving-Non-Ego-Far, where **"close" and "far" are decided by whether within the perception range of online mapping models**.

In summary, Fig. 8 illustrates the proposed evaluation agent selection strategy while Table 6 compares the motion prediction results of different groups of agents. We could conclude that (1) The difficulty of prediction is: Moving > Ego > Static. Both models perform nearly perfectly in predicting static agents' future trajectories, demonstrating the importance of excluding those from metrics. (2) The faraway agents' have worse performance compared to nearby agents, which is natural considering the missing maps.

### 3.5 Formulation of Online Map

HD Map, as an abstraction for road lines and traffic signs, could have rather different number of semantic types across different datasets Caesar et al. (2019); Wilson et al. (2021); Ettinger et al. (2021) and across different online mapping models Liao et al. (2023a;b); Li et al. (2024). For example, in nuScenes, the official semantic types include dividers, boundaries, and pedestrian crossings. In the extension of OpenLane series Chen et al. (2022); Wang et al. (2023); Li et al. (2024), centerlines, describing the virtual mid-points of lanes, and topology representing the connection relations of different map elements are introduced. In Table 5, we compare the influence of incorporating different types of online map elements for motion prediction. Not surprisingly, feeding all possible map element types into the motion prediction model leads to the best performance, as it has comprehensive information. Further, we could observe that centerlines are most helpful and centerlines only achieve the second best performance, which is natural since most people would like to drive following the center of the lane. Thus, **in OMMP-Bench we always feed all possible map elements into the motion prediction model for the best performance** while the existing framework Gu et al. (2024a) only feeds one type of map elements into the motion prediction model at each time. It could be either dividers, boundaries, or centerlines, depending on the availability of map elements in the scene.

### 4 Experiments

#### 4.1 Benchmark

In summary, Compared with existing online map based motion prediction protocol, the proposed **OMMP-Bench** has the following improvements:

1. **New Split**: we construct a new split on nuScenes dataset with three sets named the map train set, motion train set, and motion val set, which contain 367, 397, and 86 scenes respectively. With the new protocol proposed for training and evaluating online mapping based motion prediction along with the dataset, the train-val gap is eliminated. The split is carefully checked to minimize the spatial overlaps between the map train set and others, which is able to better evaluate the generalization ability of online mapping models than the original split.

2. **More Moving Agents**: We predict other agents' future trajectories which are aligned with the purpose of motion prediction while the existing one only evaluates the ego vehicle. As the motion

Table 7: **Results of Existing Online Map Based Motion Prediction Methods on Proposed OMMP-Bench**.

| Map Model | Motion Model | Method | Ego | | | Moving Non-Ego Close | | | Moving Non-Ego Far | | |
|---|---|---|---|---|---|---|---|---|---|---|---|
| | | | minADE↓ | minFDE↓ | MR↓ | minADE↓ | minFDE↓ | MR↓ | minADE↓ | minFDE↓ | MR↓ |
| MapTR | HiVT | base | 0.4015 | 0.8576 | 0.0937 | 0.5585 | 1.1476 | 0.1305 | 0.6997 | 1.3657 | 0.1854 |
| MapTR | HiVT | unc | 0.3839 | 0.8236 | 0.0956 | 0.5560 | 1.1548 | 0.1352 | 0.6946 | 1.3383 | 0.1795 |
| MapTR | HiVT | bev | 0.3812 | 0.8070 | 0.0943 | 0.5328 | 1.1184 | 0.1339 | 0.6738 | 1.3174 | 0.1772 |
| MapTR | HiVT | img | 0.3792 | 0.8032 | 0.0930 | 0.5275 | 1.1028 | 0.1332 | 0.6318 | 1.2780 | 0.1538 |
| MapTR | DenseTNT | base | 1.2114 | 2.3185 | 0.4183 | 2.0214 | 4.3053 | 0.4546 | 2.4140 | 5.0250 | 0.5055 |
| MapTR | DenseTNT | unc | 1.0486 | 2.0875 | 0.3774 | 1.8123 | 4.0554 | 0.4279 | 2.3794 | 4.5621 | 0.4994 |
| MapTR | DenseTNT | bev | 1.0856 | 2.0732 | 0.3781 | 1.8243 | 4.0384 | 0.4296 | 2.3771 | 4.6091 | 0.4991 |
| MapTR | DenseTNT | img | 0.9921 | 1.8476 | 0.3593 | 1.6851 | 3.6264 | 0.3849 | 2.0702 | 4.1723 | 0.4382 |
| MapTRv2-CL | HiVT | base | 0.3976 | 0.8571 | 0.1025 | 0.5585 | 1.1412 | 0.1301 | 0.6999 | 1.3512 | 0.1805 |
| MapTRv2-CL | HiVT | unc | 0.3862 | 0.8185 | 0.0898 | 0.5682 | 1.1751 | 0.1356 | 0.7071 | 1.3785 | 0.1856 |
| MapTRv2-CL | HiVT | bev | 0.3882 | 0.8170 | 0.0920 | 0.5632 | 1.1692 | 0.1336 | 0.7242 | 1.3944 | 0.1972 |
| MapTRv2-CL | HiVT | img | 0.3773 | 0.7991 | 0.0869 | 0.5175 | 1.0742 | 0.1320 | 0.6274 | 1.2631 | 0.1501 |
| MapTRv2-CL | DenseTNT | base | 1.1625 | 2.0731 | 0.3846 | 1.9528 | 4.1070 | 0.4284 | 2.2742 | 4.7492 | 0.4729 |
| MapTRv2-CL | DenseTNT | unc | 1.0424 | 2.0642 | 0.3570 | 1.7918 | 3.9362 | 0.4203 | 2.3666 | 4.8551 | 0.5152 |
| MapTRv2-CL | DenseTNT | bev | 1.0068 | 1.9942 | 0.3482 | 1.7738 | 3.9626 | 0.4210 | 2.3537 | 4.6741 | 0.4997 |
| MapTRv2-CL | DenseTNT | img | 0.9770 | 1.8156 | 0.3563 | 1.6482 | 3.5829 | 0.3627 | 1.9836 | 4.0002 | 0.4128 |

of stationary vehicles is easy to predict, we further propose to only evaluate moving agents and divide them into "close" and "far" groups.

3. **All Map Elements**: For the best performance, We use all possible map elements as input of the motion prediction model, while the existing benchmark only feeds one type of map elements.

4. **New Baseline**: We propose a new baseline that integrates raw image features into motion prediction models, to provide agents that are out of the range of online maps with environmental information.

### 4.2 RESULTS

The results of existing online map based motion prediction methods on OMMP-Bench are shown in Tab 7, yielding the following insights:

- **Distant agents are hard to predict**. On the one hand, across all methods, the accuracy of predicting the ego vehicle's motion is consistently better than predicting nearby vehicles, which in turn is better than predicting the motion of distant vehicles. On the other hand, methods that improve the motion prediction on ego vehicle prediction do not necessarily show similar improvements for other vehicles. For example, for several combinations of the online mapping model and motion prediction model, both MapUncertaintyPrediction Gu et al. (2024a) and MapBEVPrediction Gu et al. (2024b) methods improve ego vehicle prediction but show performance drops when predicting close non-ego agents compared to base method. When predicting far non-ego agents with the MapTRv2-CL+DenseTNT model, minADE increased by 4.1% and 4.0%. This also highlights the challenge of our proposed metrics, encouraging the online map based motion prediction model to deal with more difficult while realistic tasks.

- **Stronger online mapping model benefits motion prediction**. When using DenseTNT as the downstream module, MapTRv2-CL achieves a reduction in minADE of 4.0% relative to MapTR. Compared to MapTR, MapTRv2-CL provides higher map prediction accuracy and additionally predicts centerlines. This indicates that a stronger online mapping model can supply downstream models with richer and more accurate information, thereby enhancing their performance.

- **Integrating image feature helps to predict agents far away.** Our proposed baseline reaches a better performance on predicting farther agents. Applied the method on the MapTRv2-CL+HiVT model, the minADE decreased by 12.7%.

## 5 CONCLUSION

In this paper, we delve into the challenges and misunderstandings of the emerging online map based motion prediction protocol, which includes inappropriate data splits, different considered ranges for online mapping and motion prediction, and non-discriminative metrics. We propose the OMMP-Bench, a benchmark with new data split, refined metrics, and a new baseline as the solution. We hope OMMP-Bench could solve the misunderstanding of the new field and provide insights for further co-development of online mapping and motion prediction models.

## 6 ETHICS STATEMENT

The research conducted in the paper conforms with the ICLR Code of Ethics.

## 7 REPRODUCIBILITY STATEMENT

We describe the proposed module in Sec. 3 and detailed rules of the pipeline in Appendix A. The code and checkpoints will be open-sourced for reproduction.

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

## A    DETAILED RULES OF OMMP-BENCH

**Online Mapping Models.** On OMMP-Bench, online mapping models must be trained on the map train set. After training, the models perform inference on the motion train set and motion val set, generating and storing information required for the training and testing of motion prediction models. **There are no restrictions on the output formulations or perception range of the online mapping model**. We no not evaluate the precision of online mapping with metrics such as mAP since the output forms of different models may vary. Instead, we focus on their effectiveness in downstream motion prediction tasks, allowing flexibility in exploring the most beneficial structures and output formulations of online mapping models for motion prediction.

**Motion Prediction Models.** On OMMP-Bench, motion prediction models are trained on the motion train set and evaluated on the motion val set. Models receive the past 2 seconds of ground-truth trajectories for each agent and predict trajectories for the next 3 seconds. These trajectories are interpolated at 0.1-second intervals, with each agent having 20 points for past trajectories and 30 points for future trajectories. Models may predict up to six potential trajectories for each agent. All moving vehicles, regardless of their distance from the ego vehicle, are evaluated. We report the minADE, minFDE, and MR metrics for all moving vehicles, and categorize performance based on their location within or outside the map perception range. Motion prediction models can use any outputs or features produced by the online mapping model, such as online maps, BEV features, or image features. However, they can not use offline maps in any form.

## B    LLM USAGE

LLMs are used in writing for improving grammar and correcting typos.

