# OpenReview forum: "Understanding the Task and Data Misconceptions in Online Map Based Motion Prediction for Autonomous Driving and a Boundary-Free Baseline"
_ICLR.cc/2026/Conference — ICLR 2026 Conference Withdrawn Submission_

### Official Review · Reviewer_uhbr · 2025-10-30

**Soundness:** 3
**Presentation:** 3
**Contribution:** 2
**Rating:** 6
**Confidence:** 4

**Summary:**

This paper investigates misconceptions in online map–based motion prediction for autonomous driving, a two-stage paradigm where online mapping models generate HD maps from sensor data, which are then used for motion prediction. The authors identify key flaws in existing setups — inappropriate dataset splits, mismatched perception ranges between mapping and prediction, and non-discriminative metrics — and propose OMMP-Bench, a new benchmark addressing these issues. They also introduce a boundary-free baseline that integrates image features to compensate for missing map elements. Experiments on nuScenes show that their benchmark and baseline mitigate train–val gaps and improve motion prediction, particularly for distant agents.

**Strengths:**

1. The author design a new dataset seperation method to alleviate the train-val gap in traditional two stage motion prediction framework: stage-1 train map model on training dataset, and then stage-2 train motion prediction model using the trained map model on the same datasets. The author finds that in stage 2,, the used map model may overfit on training set and gives the motion model perfect map resulst, which maybe leads to bad test results as the map model maybe behave bad on test set. And experiment shows that model trained on new dataset seperation get better results.
2. This paper is trying to combine some experiment result into the part introducing method, I think this is a good idea for communities like autonomous driving that people care about the results of each design choice. As the reader can immediately see the experiment table after see the design choices proposed by the author.

**Weaknesses:**

Limited Strategic Relevance – The work focuses on refining a modular two-stage pipeline (mapping → prediction) while the field is shifting toward end-to-end autonomous-driving frameworks. Its contribution may have diminishing long-term impact.

Over-emphasis on Perception Range Expansion – The paper prioritizes broader spatial coverage as a performance driver, overlooking the fact that practical systems and human drivers focus on risk-relevant objects rather than exhaustive detection.

**Questions:**

1. In line 321-322, you said that "To enable those distant agents to obtain environment information, we propose a new baseline in OMMP-Bench to allow all agents to extract features from their corresponding nearby regions of raw image features." So, do you use the image produced by camera munted on ego car? If so, those agents far away from the ego seems likely to be obscured by other agents more close to ego car.
2. In your framework, you found that integrating image feature helps to predict agents far away, do you think "integrateing image features (maybe from perception module)" is a kind of design choice of "end to end" (instead of set planning as last goal, here motion prediction is the final goal)?


Typos:
At line 230-231, ignored space after ③

---

### Official Review · Reviewer_K2Xt · 2025-10-31

**Soundness:** 2
**Presentation:** 1
**Contribution:** 1
**Rating:** 2
**Confidence:** 4

**Summary:**

This paper points out that there exists an hd map construction performance train-val gap in the trajectory prediction task based on online hd map, resulting in the trajectory prediction part being trained on high-quality online hd map but tested on low-quality online hd map. Therefore, nuScenes is divided into three parts: Map train, Motion train, and Motion val to solve this problem.

**Strengths:**

1.Motivation is clear

**Weaknesses:**

1.Unfortunately, I do not consider the "train-val gap" proposed in this paper to be a problem. In practical commercial applications of autonomous driving, there are usually separate training data for map generation and motion plan, which do not overlap, so this problem does not exist.
2.The non-overlapping division method of nuScenes proposed in this paper has already been presented in StreamMapNet.
3.The approach of integrating image features into the trajectory prediction process proposed in this paper is the same as that in ViP3D, where the 3D query in ViP3D includes 2D features from images.
4.As a submission to ICLR 2026, this paper actually does not discuss any relevant articles published in 2025. I believe the authors have overlooked many of the latest advancements.
5.Without the open-source nuscenes division of this paper, I cannot even verify the authenticity of the experiments, as the experiments on the baseline methods in this paper are not conducted on the previously published nuscenes division.

**Questions:**

What are the data volumes of different splits between the nuScenes division in this paper and the default nuScenes division?

---

### Official Review · Reviewer_NpAT · 2025-11-05

**Soundness:** 3
**Presentation:** 2
**Contribution:** 3
**Rating:** 4
**Confidence:** 3

**Summary:**

This paper studies the online map-based motion prediction (OMMP) paradigm and argues that current protocols are flawed due to (1) train–val gaps in dataset splits, (2) spatial range mismatch between online map estimation and motion prediction, and (3) metrics that overemphasize trivial ego/static agents. The authors propose OMMP-Bench, a benchmark with spatially disjoint splits and evaluation focused on non-ego moving agents, and introduce a boundary-free baseline that supplements map features with deformable-attention image features to provide context for agents outside the map coverage range. Experiments across multiple mapping and prediction models show that the revised protocol yields more discriminative evaluations and that the baseline improves far-agent accuracy.

**Strengths:**

- Clearly identifies and empirically demonstrates key issues in existing OMMP evaluation setups.
- Provides a benchmark with improved data splits and metrics that better reflect the intended prediction task.
- Introduces a simple and practical baseline that addresses the map coverage mismatch and yields consistent improvements for far-away agents.
- Conducts comprehensive evaluations across multiple combinations of mapping and prediction models.

**Weaknesses:**

- The primary contribution is benchmark and protocol correction, with limited methodological novelty.
- The boundary-free baseline is incremental, given that deformable attention feature aggregation is already widely used in recent end-to-end forecasting and driving frameworks (UniAD, SparseDrive).
- Limited discussion on how OMMP-Bench insights transfer to end-to-end pipelines where mapping and prediction are co-trained.
- All experiments rely on ground-truth agent trajectories. Robustness to upstream detection/tracking noise is not evaluated, which limits conclusions about real-world deployment.

**Questions:**

- How sensitive are results to the specific geographic split design?
- How does performance compare if deformable-attention image features are used for all agents (i.e., map-free baseline)?
- Does the improved data split also benefit end-to-end visual forecasting pipelines (e.g., UniAD, SparseDrive)?
- What is the runtime and memory overhead of the boundary-free baseline and how does this compare to previous approaches?
- How robust are the evaluated methods to noise in agent positions and histories?

---

### Note · Authors · 2025-11-12

I have read and agree with the venue's withdrawal policy on behalf of myself and my co-authors.